# The optimal duration of high-intensity static stretching in hamstrings

**Kosuke Takeuchi**[1]*, **Masatoshi Nakamura**[2]

**1** Department of Physical Therapy, Kobe International University, Kobe, Hyogo, Japan, **2** Institute for Human Movement and Medical Sciences, Niigata University of Health and Welfare, Niigata, Niigata, Japan

* ktakeuchi@kobe-kiu.ac.jp

**Data Availability Statement:** All relevant data are within the manuscript and its Supporting Information files.

**Funding:** The authors received no specific funding for this work.

## Abstract

### Objectives

The purpose of this study was to compare the duration of high-intensity static stretching on flexibility and strength in the hamstrings.

### Methods

Fourteen healthy males (20.8 ± 0.6 years, 170.7 ± 6.5 cm, 66.4 ± 9.9 kg) underwent high-intensity static stretching for three different durations (10, 15, and 20 seconds). The intensity of static stretching was set at the maximum point of discomfort. To examine the change in flexibility and strength, range of motion, peak passive torque, relative passive torque, muscle-tendon unit stiffness, peak torque of isokinetic knee flexion, and knee angle at peak torque of isokinetic knee flexion were measured. To evaluate a time course of pain, a numerical rating scale was described.

### Results

Range of motion ($P < 0.01$), peak passive torque ($P < 0.01$), and knee angle at peak torque were increased at all interventions. Relative passive torque ($P < 0.01$) and muscle-tendon unit stiffness ($P < 0.01$) were decreased at all interventions. Peak torque decreased after 10 seconds of stretching ($P < 0.05$). Numerical rating scale during stretching was 8–9 levels in all interventions, the pain disappeared immediately after the post-measurements (median = 0).

### Conclusion

The results suggested that muscle-tendon unit stiffness decreased regardless of duration of high-intensity static stretching. However, peak torque of isokinetic knee flexion decreased after 10 seconds of high-intensity static stretching, though it was no change after for more than 15 seconds of stretching.

**Competing interests:** The authors have declared that no competing interests exist.

## Introduction

Static stretching (SS) is commonly used as a part of a warm-up routine in order to increase range of motion (ROM) and potentially prevent injuries [1]. Previous review studies reported that SS increases ROM effectively [2–5]. However, it is pointed out that the passive property of the muscle-tendon unit and tolerance for stretching influences ROM [5–7]. The passive property of the muscle-tendon unit is measured by using muscle-tendon unit stiffness, which is calculated from the slope of the torque-angle curve during passive joint movement [8–10]. On the other hand, tolerance for stretching is measured by peak passive torque during passive joint movement [10–12]. ROM is defined as one at which the subject is able to have a maximally tolerable angle without pain. An increase in the passive torque during the passive joint movement indicated that the subjects received higher force without pain, which indicates that an increase of tolerance is obtained.

Previous studies reported that muscle-tendon unit stiffness is related to the occurrence of muscle-tendon injuries [13–15]. Witvrouw et al. [16] suggested that a sufficient level of muscle-tendon unit compliance is needed for sports utilizing a stretch-shortening cycle to effectively store and release a high amount of elastic energy. In the case of insufficient muscle-tendon unit compliance, the demands in energy absorption and release may rapidly exceed the capacity of the muscle-tendon unit, which may cause a higher risk of injuries [6,17,18]. Many previous studies reported that SS decrease muscle-tendon unit stiffness effectively [7,19,20]. Therefore, SS used as a part of warm-up routine decreases muscle-tendon unit stiffness and could lead to preventing injuries.

The effects of SS on muscle-tendon unit stiffness is affected by its duration [9,21] and intensity [7,8,22]. Matsuo et al. [9] and Nakamura et al. [21] examined the duration of SS needed to decrease muscle-tendon unit stiffness of hamstrings, and they showed that three minutes of SS was needed to decrease the stiffness. However, previous studies reported the duration of SS used as a part of a warm-up routine is approximately 20 seconds [1,23]. In fact, Takeuchi et al. [22] reported that 20 seconds of SS in hamstrings does not change muscle-tendon unit stiffness. Moreover, Sato et al. [24] examined the influence of SS for 20 seconds in gastrocnemius muscle on shear elastic modulus, which is an indicator as a passive property of the muscle measured by using shear wave elastography, and showed the shear elastic modulus did not change after the stretching. These data suggest that SS for 20 seconds used as a part of a warm-up routine does not change muscle-tendon unit stiffness and may not be able to prevent injury as well. However, because many athletes practice within a limited time, it is difficult to perform SS for more than 180 seconds for each muscle. Therefore, it is necessary to develop an SS that can decrease muscle-tendon unit stiffness in 20 seconds.

The influence of SS is affected by its intensity as well as its duration [8,22]. Kataura et al. [8] examined the effect of different intensities of SS and reported that there was a significant moderate negative correlation between the intensity of SS and relative change in passive stiffness. Moreover, our previous study examined high-intensity of SS for 20 seconds in hamstrings and showed that muscle-tendon unit stiffness decreased after the stretching [22]. However, SS at high intensity is accompanied by moderate to severe pain [8,22]. Our previous study measured the pain level by using a numerical rating scale [NRS; 11-point scale raged from 0 (no pain) to 10 (worst imaginable pain)] [22]. The results indicated that median of NRS during SS at high intensity was 8, although NRS immediately after the stretching and 24 hours after the stretching were both 0 levels [22]. To minimize the pain during SS at high intensity, it is important to examine the optimal duration of the stretching.

A previous study investigating static stretching protocol reported that SS for less than 10 seconds was rarely performed [1]. On the other hand, SS for 10 to 19 seconds was frequently

used [1]. Therefore, the purpose of this study was to compare the duration of high-intensity SS (10, 15, and 20 seconds) on flexibility and strength in the hamstrings and to clarify the optimal duration of SS at high intensity.

## Materials and methods

### Participants

Fourteen healthy men (20.8 ± 0.6 years, 170.7 ± 6.5 cm, 66.4 ± 9.9 kg) were recruited. Participants who regularly performed any flexibility and strength training or who had a history of lower limb pathology were excluded. The sample size was calculated with a power of 80%, alpha error of 0.05, and effect size of 0.25 (middle) using G*Power 3.1 software (Heinrich Heine University, Düsseldorf, Germany), and the results showed that the requisite number of participants for this study was 14. All participants were informed of the requirements and risks associated with their involvement in this study and signed a written informed consent document. The study was performed in accordance with the Declaration of Helsinki (1964). The Ethics Committee of Kobe International University approved the study.

### Procedure

The purpose of the present study was to examine the optimal duration of high-intensity SS in hamstrings. For this purpose, the participants underwent three different duration of high-intensity SS (10 sec, 15 sec, and 20 sec) in the right hamstrings, in random order. The participants visited three times on a separate day, with an interval of one week. Participants attended a familiarization session at least 24 hours before the first testing day. To evaluate any alteration of flexibility and strength of the right hamstrings, ROM, passive torque, peak torque during isokinetic knee flexion and knee angle at peak torque of isokinetic knee flexion were measured before and after each SS. In addition, NRS was examined during SS, immediately after the post-measurement, and 24 hours after SS. The experiment was performed in a university laboratory, where the temperature was maintained at 25 degrees C.

### Flexibility assessment

The flexibility assessment was performed in the same fashion as a previous study [8,22]. The previous study reported that the reliability of the measurements used in this study was acceptable [8]. An isokinetic dynamometer machine (CYBEX NORM, Humac, California, USA) was used in the present study. This study used a sitting position in which the hip joint was flexed, which has been shown to efficiently stretch the hamstrings [8]. The participants were seated on a chair with the seat tilted maximally, and a wedge-shaped cushion was inserted between the trunk and the backrest, which set the angle between the seat and the back at approximately 60 degrees. The previous study, which used the same assessment, reported that the average angle of hip flexion was 111.2 ± 2.5 degrees [8]. The chest, pelvis, and right thigh were stabilized with straps. The right knee joint was aligned with the axis of the rotation of the isokinetic dynamometer machine. The lever arm attachment was placed just proximal to the malleolus medialis and stabilized with straps. In the present study, reported knee angles were measured using the isokinetic dynamometer machine. A 90-degree angle between the lever arm and floor was defined as 0 degrees of knee flexion/extension. The participants were instructed to relax during the flexibility assessment.

### ROM

The knee joint was passively extended from 0 degrees to maximum angle without pain at 5 degrees/seconds. A previous study showed that the velocity does not cause stretch reflex [25]. ROM was defined as the range from 0 degrees to the maximum knee extension angle.

### Passive torque

The passive torque during ROM measurement was recorded in the isokinetic dynamometer machine. Passive torque was obtained at two points [11]. (1) relative passive torque was obtained at the same knee angle reached before SS to examine changes in passive properties of the muscle-tendon unit. (2) Peak passive torque was obtained at maximum ROM both before and after SS to examine tolerance for stretching.

### Calculation of muscle-tendon unit stiffness

Muscle-tendon unit stiffness was defined as the values of the slope of the regression line that was calculated from the torque-angle relationship using the least-squares method [10]. Muscle-tendon unit stiffness was calculated from the same knee extension angle range before and after SS. The calculated knee extension angle range was defined as the angle from the 50% maximum knee extension angle to the maximum knee extension angle measured before SS [8,9,22].

### Peak torque and knee angle during maximum voluntary isokinetic knee flexion

The peak torque of knee flexion during maximum voluntary isokinetic knee flexion at 60 degrees/second was measured. The participants were secured on the isokinetic dynamometer machine in the same fashion as the measurement of ROM. The range of the measurement was set from 0 degrees to maximum knee extension angle. The participants performed three sub-maximal isokinetic knee flexion as a warm-up trial. After the warm-up trial, the participants performed three maximum isokinetic knee flexion as muscle strength measurement. The greatest values of three isokinetic knee flexion were used for the analyses as the peak torque. The joint angle at the peak torque was provided by the isokinetic dynamometer machine.

### Numerical rating scale

The level of pain during SS, immediately after the post-measurement, and 24 hours after SS were quantified by an 11-point NRS that ranged from 0 (no pain) to 10 (worst imaginable pain) [8,22].

### Static stretching

SS was performed on the isokinetic dynamometer machine in the same fashion as the measurement of ROM. The knee joint was passively extended from 0 degrees to the angle at the maximum point of discomfort [22]. The angle was held for three different duration (10 sec, 15 sec, and 20 sec). Thereafter, the knee joint was passively returned to 0 degrees. The participants received one set of stretching at each different duration. The participants were instructed to relax during each stretch. The percent change in angle during SS respect to ROM before the stretching was defined as the intensity of the stretching [8,22].

## Statistical analyses

All variables except NRS were described as mean ± SD in the present study. NRS was described as a median. A one-way repeated measure analysis of variance was used to examine the difference in intensity of SS. For variables except for NRS, a two-way repeated measures analysis of variance was used to examine the effects of intervention (10 sec vs. 15 sec vs. 20 sec) and time (pre vs. post). For NRS, a two-way repeated-measures analysis of variance was used to examine the effects of intervention (10 sec vs. 15 sec vs. 20 sec) and time (during SS vs. immediately after the post-measurements vs. 24 hours after SS). If a significance was detected, post hoc analyses using Bonferroni's test were performed. Spearman's rank correlation coefficient was conducted between the intensity of SS and relative change of ROM, relative passive torque, peak passive torque, and muscle-tendon unit stiffness. The analyses were performed using SPSS version 25 (SPSS, Inc., Chicago, IL, USA). Differences were considered statistically significant at an alpha level of $p < 0.05$. To describe the effect size, the partial eta squared value was calculated by using the SPSS software.

## Results

### Intensity of SS

The intensity of SS for each intervention were as follows: 10 sec, 137.9 ± 11.4%; 15 sec, 128.9 ± 12.7%; 20 sec, 130.3 ± 8.6%. There was no significant difference in intensity of SS between interventions ($p = 0.07$, partial eta squared = 0.19).

### ROM

For ROM, there was no significant two-way interaction (intervention × time, $p = 0.12$, partial eta squared = 0.10) and no main effect for intervention ($p = 0.37$, partial eta squared = 0.05), but there was significant main effect for time ($p < 0.01$, partial eta squared = 0.75) (Fig 1). ROM increased after the stretching ($p < 0.01$).

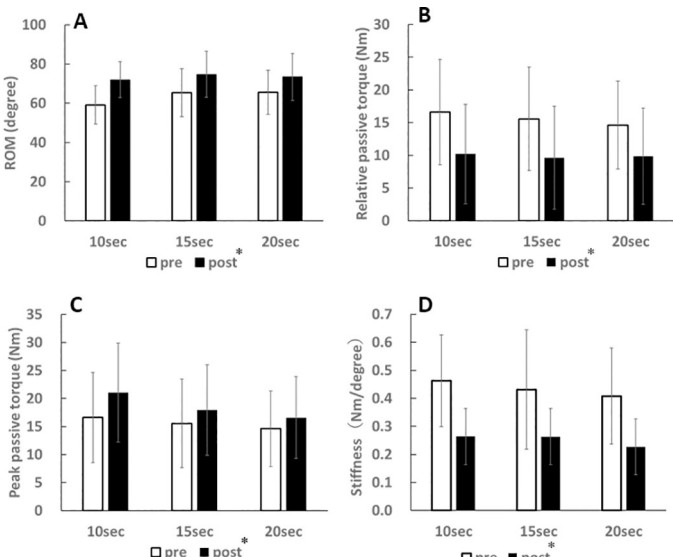

**Fig 1. Effects of stretching on changes in ROM (A), relative passive torque (B), peak passive torque (C), and muscle-tendon unit stiffness (D).** *$p < 0.01$ (pre vs. post).

### Relative passive torque

For relative passive torque, there was no significant two-way interaction (intervention × time, $p = 0.63$, partial eta squared = 0.02) and no main effect for intervention ($p = 0.84$, partial eta squared = 0.01), but there was significant main effect for time ($p < 0.01$, partial eta squared = 0.86) (Fig 1). Relative passive torque decreased after each SS ($p < 0.01$).

### Peak passive torque

For peak passive torque, there was no significant two-way interaction (intervention × time, $p = 0.42$, partial eta squared = 0.04) and no main effect for intervention ($p = 0.49$, partial eta squared = 0.04), but there was significant main effect for time ($p < 0.01$, partial eta squared = 0.22) (Fig 1). Peak passive torque increased after the stretching ($p < 0.01$).

### Muscle-tendon unit stiffness

For muscle-tendon unit stiffness, there was no significant two-way interaction (intervention × time, $p = 0.62$, partial eta squared = 0.02) and no main effect for intervention ($p = 0.78$, partial eta squared = 0.01), but there was significant main effect for time ($p < 0.01$, partial eta squared = 0.84) (Fig 1). Muscle-tendon unit stiffness decreased after the stretching ($p < 0.01$).

### Peak torque of isokinetic knee flexion

For peak torque of isokinetic knee flexion, there was significant two-way interaction (intervention × time, $p < 0.05$, partial eta squared = 0.15) (Fig 2). Post hoc analysis revealed that the peak torque decreased after 10 sec of SS ($p < 0.05$) while there was no change after 15 and 20 sec of SS ($p = 0.96$, $0.30$, respectively).

### Knee angle at peak torque of isokinetic knee flexion

For knee angle at peak torque of isokinetic knee flexion, there was no significant two-way interaction (intervention × time, $p = 0.42$, partial eta squared = 0.04) and no main effect for intervention ($p = 0.52$, partial eta squared = 0.03), but there was significant main effect for time ($p < 0.01$, partial eta squared = 0.52) (Fig 2). The knee angle increased after the stretching ($p < 0.01$).

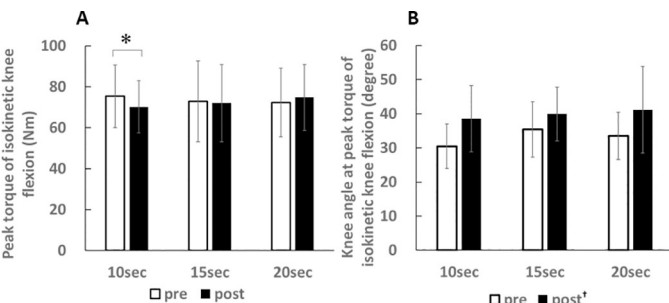

**Fig 2. Effects of stretching on changes in peak torque of isokinetic knee flexion (A) and knee angle at peak torque of isokinetic knee flexion (B).** $^{*}p < 0.05$ (pre vs. post). $^{\dagger}p < 0.01$ (pre vs. post).

### NRS

For NRS, there was no significant two-way interaction (intervention × time, p = 0.54, partial eta squared = 0.04) and no main effect for intervention (p = 0.72, partial eta squared = 0.02), but there was significant main effect for time (p < 0.01, partial eta squared = 0.98). NRS during SS [median values (interquartile range):10 sec = 9 (8–9), 15 sec = 8 (8–9), 20 sec = 8 (8–9)] indicated higher values compared with those immediately after the post-measurements [median values (interquartile range): 0 (0 = 0) in all interventions] (p < 0.01) and 24 hours after SS [median values (interquartile range): 0 (0 = 0) in all interventions] (p < 0.01).

### Correlation between intensity and measurements of flexibility

There was significant correlations between intensity of SS and the relative change in ROM (10 sec, r = 0.88, p < 0.01; 15 sec, r = 0.94, p < 0.01; 20 sec; r = 0.52, p < 0.05), relative passive torque (10 sec, r = -0.62, p < 0.05; 15 sec, r = -0.78, p < 0.01; 20 sec, r = -0.72, p < 0.01), and muscle-tendon unit stiffness (10 sec, r = -0.61, p < 0.05; 15 sec, r = -0.74, p < 0.01; 20 sec, r = -0.62, p < 0.05). On the other hand, there was no significant correlations between intensity of SS and relative change in peak passive torque (10 sec, r = 0.51, p = 0.07; 15 sec, r = 0.29, p = 0.32; 20 sec, r = 0.14, p = 0.63).

## Discussion

In this study, SS was performed at the angle where the maximum point of discomfort. The results showed that there was no significant difference in the intensity of SS between interventions. In addition, there was no significant difference in NRR during SS between interventions. These results indicated that the participants received SS at similar intensity subjectively and objectively regardless of the duration of SS.

In the present study, regardless of the interventions, ROM and peak passive torque significantly increased but relative passive torque and muscle-tendon unit stiffness significantly decreased. Alteration in ROM after SS are attributed to alterations in passive property of the muscle and tolerance for stretching [5–7]. In the present study, relative passive torque [11] and muscle-tendon unit stiffness [7,10,19] were measured as indicators of the passive property of the hamstrings, and peak passive torque was measured as an indicator of tolerance for stretching [10,26,27]. Previous studies showed that ROM and peak passive torque increased, but muscle-tendon unit stiffness of the hamstrings decreased after high-intensity SS for 20 [22] and 180 seconds [8]. These results suggested that the effects of the duration of high-intensity SS was small, and the stretching increased ROM through a decrease in relative passive torque and muscle-tendon unit stiffness and an increase in tolerance for stretching.

In all interventions, there were significant correlations between the intensity of SS and relative changes in ROM, relative passive torque, and muscle-tendon unit stiffness. However, there were no significant correlations between the intensity of SS and peak passive torque. Our previous study examined the effects of high-intensity SS for 20 seconds and showed that there were significant correlations between the intensity of SS and ROM, muscle-tendon unit stiffness, and peak passive torque [22]. While, Kataura et al. [8] examined the effect of high intensity of SS for 180 seconds in hamstrings and reported that there were significant correlations between the intensity of SS and ROM and passive stiffness, though there was no significant correlation between the intensity of SS and peak passive torque. From these three studies, when the high-intensity SS is performed, it is suggested that the intensity of SS is important for the change in ROM and passive property of the hamstrings. On the other hand, no consensus has been obtained regarding the relationship between the intensity of SS and changes in tolerance for stretching. Although the details of the mechanisms for the increase in tolerance for

stretching are unclear, previous studies have reported that the increase in stretch tolerance was attributed to decrement in the perceptions of pain and discomfort accompanied by a change of neural and psychological factors after stretching [28,29]. Previous studies reported that change in ROM after SS at the intensity of no pain is more affected by stretch tolerance than passive properties [30,31]. The change in tolerance for stretching after high-intensity SS needs to be studied in more detail.

In the present study, the peak torque of isokinetic knee flexion was decreased after SS for 10 seconds, while it was no change after SS for 15 and 20 seconds. In our previous study, the peak torque was not changed after high-intensity SS for 20 seconds [22], and in this regard, the results of the present study support the previous study. The alteration in muscle strength after SS is attributed to an alteration in muscle-tendon unit stiffness [32,33] and neural activity [34–37]. SS theoretically decreases the force transfer efficiency from the muscle to the skeleton [38] with the decrement in muscle-tendon unit stiffness [32,33] and rightward shift of torque-angle curve [39–41]. Trajano et al. [35] reported that central factors were strongly related to the torque reduction immediately after SS and during torque recovery. In the present study, decrement in muscle-tendon stiffness and increment in knee angle at peak torque of isokinetic knee flexion were found regardless of its duration. Increment in the knee angle at peak torque indicated a rightward shift of torque angle curve [41]. Therefore, it is suggested that the difference in peak torque of isokinetic knee flexion was caused by neural activity because there were no differences in changes in muscle-tendon unit stiffness and knee angle at peak torque between interventions.

Previous studies reported that sympathetic nerve activity is activated following pain and discomfort levels [42,43]. The sympathetic nerve plays a crucial role in blood flow and neural activity during muscle contraction [44]. In the present study, NRS during SS was 8–9 levels in all interventions, which indicated that the participants felt severe pain. To our best knowledge, it has not been examined to the extent of stretching stimulation required for sympathetic nerve excitation. However, it is possible that 10 seconds of high-intensity SS was insufficient for sympathetic nerve excitation, and the sympathetic nerve excitation required for more than 15 seconds of the stretching. Therefore, it is suggested that the peak torque of isokinetic knee flexion was decreased because of the decrement in muscle-tendon unit stiffness after 10 seconds of high-intensity SS. while the effect of the decrement in the stiffness would be offset by an increment in sympathetic nerve activity after 15 and 20 seconds of the stretching. However, the present study did not measure sympathetic nerve activity. In addition, EMG activity of the hamstrings could not be measured because the right thigh was stabilized to the seat with a strap in order to accurately perform measurements and interventions. It is necessary to examine the changes in neural activity after high-intensity SS in detail in order to clarify the mechanism of change in the peak torque of isokinetic knee flexion after the stretching.

In the present study, NRS was 8–9 levels during SS, which indicated that participants felt severe pain. However, the pain was disappeared immediately after the post-measurements, and there was no pain 24 hours after SS. These results are consistent with our previous study [22]. These data suggested that risk of high-intensity SS may be low for healthy men. However, Apostolopoulos [45] compared the effects of high- (70%-80% maximum perceived stretch) and low-intensity (30%-40% maximum perceived stretch) SS on recovery from unaccustomed eccentric exercise. They showed that low-intensity stretching is likely to result in small-to-moderate beneficial effects on perceived muscle soreness and recovery of muscle function post-unaccustomed eccentric exercise [45]. Therefore, it is necessary to investigate the safety of high-intensity SS for the persons with muscle-tendon disorders.

In summary, the results showed that muscle-tendon unit stiffness decreased regardless of its duration. However, peak torque of isokinetic knee flexion, which is an indicator of muscle

strength of the hamstrings, decreased after high-intensity SS for 10 seconds, although there was no change after more than 15 seconds of high-intensity SS. The decrement in muscle strength after SS is restored within 10 minutes [33]. Moreover, performing activities (jumps and sprints) following SS can mitigate the negative effects of SS [46,47]. These data suggested that if athletes elect to stretch statically, they need to choose the duration of high-intensity SS taking into account their subsequent activities. Athletes who need great muscle strength immediately after high-intensity SS without any activities should use more than 15 seconds of stretching. If athletes have more than 10 minutes or perform activities after high-intensity SS, they should use 10 seconds of high-intensity SS to minimize the pain.

## Conclusions

The present study showed that, regardless of the duration of SS, ROM and peak passive torque increased, but relative passive torque and muscle-tendon unit stiffness decreased. Peak torque of isokinetic knee flexion decreased after 10 seconds of high-intensity SS. High-intensity SS for 10 seconds could be useful to increase flexibility and prevent injuries. However, athletes should use high-intensity SS for more than 15 seconds when they need great muscle strength immediately after stretching.

## Supporting information

**S1 File.**
(XLSX)

## Author Contributions

**Conceptualization:** Kosuke Takeuchi.

**Data curation:** Kosuke Takeuchi.

**Investigation:** Kosuke Takeuchi, Masatoshi Nakamura.

**Methodology:** Kosuke Takeuchi, Masatoshi Nakamura.

**Project administration:** Kosuke Takeuchi.

**Resources:** Kosuke Takeuchi.

**Software:** Kosuke Takeuchi.

**Visualization:** Kosuke Takeuchi.

**Writing – original draft:** Kosuke Takeuchi.

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
