## [Decision Letter · Decision Letter 0]

17 Aug 2020

PONE-D-20-18359

The optimal duration of high-intensity static stretching in hamstrings

PLOS ONE

Dear Dr. Takeuchi,

Thank you for submitting your manuscript to PLOS ONE. After careful consideration, we feel that it has merit but does not fully meet PLOS ONE’s publication criteria as it currently stands. Therefore, we invite you to submit a revised version of the manuscript that addresses the points raised during the review process.

The reviewers have made several comments focussing on the clarity of expression of this manuscript. Please address these, particularly the need for Table 1 and the numerous spelling and grammatical errors throughout the manuscript.

We look forward to receiving your revised manuscript.

Kind regards,

Andrew Philip Lavender, PhD

Academic Editor

PLOS ONE

Journal Requirements:

Reviewers' comments:

Reviewer's Responses to Questions

**Comments to the Author**

1. Is the manuscript technically sound, and do the data support the conclusions?

Reviewer #1: Yes

Reviewer #2: Yes

2. Has the statistical analysis been performed appropriately and rigorously? 

Reviewer #1: Yes

Reviewer #2: Yes

3. Have the authors made all data underlying the findings in their manuscript fully available?

Reviewer #1: No

Reviewer #2: Yes

4. Is the manuscript presented in an intelligible fashion and written in standard English?

Reviewer #1: No

Reviewer #2: Yes

5. Review Comments to the Author

Reviewer #1: Thank you for the opportunity to review this manuscript. I commend the authors for undertaking this research project.

General comments: A concern that I have is of a practical nature. Static stretching is rarely ever performed in isolation prior to competing in sport or participating in exercise. It is known that high volume static stretching can dampen muscular strength/power performance. However, it is also known that performing an activity (jumps and sprints) specific warm-up following a static stretching protocol can mitigate the negative consequences of the static stretch (Samson et al. 2012 - Journal of Sports Science and Medicine; Young et al. 2006 J sports med and physical fitness). I question whether the practitioners as end users of this manuscript will be able to utilize these findings to inform their practice.

Specific Comments:

Abstract:

Line 17- 'the' hamstrings

Line 20 - change men to 'males'

Line 29 - there is an extra space between torque and (P<0.01)

Line 34 - there appears to be a subheading 'interpretation' with no information following it

Introduction:

General - there are numerous missing words. I have done my best to indicate where and what they are but may have missed some.

Line 44 - The effectiveness of static stretching for injury prevention is debatable it is recommended that the authors change wording to say "potentially" prevent injuries.

Line 45 - change to increases and att 'the' prior to muscle-tendon unit.

Line 46 - change to influences ROM.

Line 48- add 'the' prior torqu anngle curve

Line 53 - conducted should be changed to 'utilising'

Line 54 - in 'the' case off

Line 62 - the number 3 should be spelled out in full

Line 72 - Do we always want to decrease stiffness? For activities such as jumping, sprinting and changing directions stiffness is beneficial.

Methods:

General- Unclear howm many repetitions of each static stretch were undertaken? The authors do not mention the reliability of these measurements.

Specific

Line 174 - 'Analysis' of variance

Statistical analysis:

No mention of the calculation of effects via partial eta squared however they are presented in the results section.

No mention of correlations was made in this section however correlational results are presented (Lines 249-256)

Conclusions:

There could be more information surrounding the practical implications of the results.

Reviewer #2: The paper is clearly written, and I found it easy to follow for the most part. There are some minor grammar or editing errors to correct, and some more clarity needed in the methods.

Introduction:

• Line 45: “increases”.

• Line 45: “that the passive property of the muscle-tendon unit”…

• Lines 46-50: It’s not clear to me what the difference in the measurements reported here is. Please clarify.

• The reference used to justify the link between muscle-tendon unit stiffness, and injury risk, is quite dated. In the last 20 years there has been a significant body of research that examines static stretching and injury risk, which seems to have been ignored here. It is important that this is addressed in the introduction.

• Line 59: delete “it”

• Line 69: should read “did not change” or “was not changed”

• Line 83: I’m not sure that “demerit” is the word the authors are looking for here, but it's not clear to me what is meant.

Materials and Methods:

• The authors make a point of specifying that 20 sec was the stretching duration they wanted to assess. And this is the first mention of 10 sec and 15 sec. A justification for these durations in the introduction would be helpful.

• Mostly methods were very well reported. However, I’m a little unclear about how the stretch was obtained. What was the hip angle during stretching and testing? This will have a significant impact on the stretch that is felt at different angles of knee extension, but I can’t see this reported anywhere.

• Line 174: “one-way repeated measures analysis of variance”

• The data analysis was very clear and detailed.

Results:

• I’m not sure about the need for Table 1. This is hardly a key result, and is not very informative, so this can be reported in the text. I’m not even sure the statistical analysis reported is necessary here, other than to compare the conditions during stretching.

Discussion:

• Paragraph starting Line 294 needs to be broken up. There are a number of ideas here, and this is hard to follow.

• Line 320: is a word missing here?

• Line 334: “person”

• Line 338: is “fore” an extra work here?

6. PLOS authors have the option to publish the peer review history of their article (what does this mean?). If published, this will include your full peer review and any attached files.

Reviewer #1: **Yes: **Scott Talpey

Reviewer #2: **Yes: **Daniel Jolley

---

## [Author Response · Author response to Decision Letter 0]

28 Aug 2020

Response to reviewers:

We wish to express our strong appreciation to the Reviewers for their insightful comments on our paper. In this response letter, the line numbers for the manuscript labeled “Revised Manuscript with Track change” are shown.

Response to Reviewer #1

Comment 1:

General comments: A concern that I have is of a practical nature. Static stretching is rarely ever performed in isolation prior to competing in sport or participating in exercise. It is known that high volume static stretching can dampen muscular strength/power performance. However, it is also known that performing an activity (jumps and sprints) specific warm-up following a static stretching protocol can mitigate the negative consequences of the static stretch (Samson et al. 2012 - Journal of Sports Science and Medicine; Young et al. 2006 J sports med and physical fitness). I question whether the practitioners as end users of this manuscript will be able to utilize these findings to inform their practice.

Response 1:

Thank you for the comment. As you indicated, performing activities can mitigate the negative effects of static stretching such as decrement in muscle strength (Young et al. 2006 J sports med and physical fitness). Therefore, it is possible that a decrement in muscle strength after 10 seconds of high-intensity static stretching can be offset by performing activities after the stretching. We considered that the point was important for the practitioners to utilize our findings in their practice. We changed the text as follows.

From

“In summary, the results showed that muscle-tendon unit stiffness decreased regardless of its duration. However, peak torque of isokinetic knee flexion, which is an indicator of muscle strength of hamstrings, decreased after high-intensity SS for 10 seconds, though it was no change after fore more than 15 seconds of high-intensity SS. The decrement in muscle strength after SS is restored within 10 minutes [31]. These data suggested that athletes should need to choose the duration of high-intensity SS taking into account their subsequent activities. Athletes who need great muscle strength immediately after high-intensity SS should use for more than 15 seconds stretching. If there is more than 10 minutes after high-intensity SS, athletes should use for 10 seconds of high-intensity SS to minimize the pain.”

To

“In summary, the results showed that muscle-tendon unit stiffness decreased regardless of its duration. However, peak torque of isokinetic knee flexion, which is an indicator of muscle strength of the hamstrings, decreased after high-intensity SS for 10 seconds, although there was no change after more than 15 seconds of high-intensity SS. The decrement in muscle strength after SS is restored within 10 minutes [31]. Moreover, performing activities (jumps and sprints) following SS can mitigate the negative effects of SS [46,47]. These data suggested that if athletes elect to stretch statically, they need to choose the duration of high-intensity SS taking into account their subsequent activities. Athletes who need great muscle strength immediately after high-intensity SS without any activities should use more than 15 seconds of stretching. If athletes have more than 10 minutes or perform activities after high-intensity SS, they should use 10 seconds of high-intensity SS to minimize the pain.” (Line 352-362)

Comment 2: 

Abstract:

Line 17- 'the' hamstrings

Response 2: 

Thank you for the comment. We revised it. (Line 17)

Comment 3: 

Line 20 - change men to 'males'

Response 3: 

Thank you for the comment. We revised it. (Line 20)

Comment 4: 

Line 29 - there is an extra space between torque and (P<0.01)

Response 4: 

Thank you for the comment. We revised it. (Line 29)

Comment 5: 

Line 34 - there appears to be a subheading 'interpretation' with no information following it

Response 5: 

Thank you for the comment. The subheading was unnecessary. We omitted it. (Line 34)

Comment 6: 

Line 44 - The effectiveness of static stretching for injury prevention is debatable it is recommended that the authors change wording to say "potentially" prevent injuries.

Response 6: 

Thank you for the comment. As you indicated, the effectiveness of static stretching for injury prevention is debatable. Therefore, we revised the text in accordance with your comment. (Line 44)

Comment 7: 

Line 45 - change to increases and att 'the' prior to muscle-tendon unit.

Response 7: 

Thank you for the comment. We revised it. (Line 45)

Comment 8: 

Line 46 - change to influences ROM.

Response 8: 

Thank you for the comment. We revised it. (Line 46)

Comment 9: 

Line 48- add 'the' prior torqu anngle curve

Response 9: 

Thank you for the comment. We revised it. (Line 48)

Comment 10: 

Line 53 - conducted should be changed to 'utilizing'

Response 10: 

Thank you for the comment. We revised it. (Line 56)

Comment 11: 

Line 54 - in 'the' case off

Response 11: 

Thank you for the comment. We revised it. (Line 57)

Comment 12: 

Line 62 - the number 3 should be spelled out in full

Response 12: 

Thank you for the comment. We revised it. (Line65)

Comment 13: 

Line 72 - Do we always want to decrease stiffness? For activities such as jumping, sprinting and changing directions stiffness is beneficial.

Response 13: 

Thank you for the comment. The main purposes of the static stretching are to increase flexibility and to prevent injuries (Kosuke Takeuchi, International Journal of Sport and Health Science, 2019). From the viewpoint of achieving the purposes of the static stretching, decrement in muscle-tendon unit stiffness is important. However, static stretching theoretically decreases the force transfer efﬁciency from the muscle to the skeleton and causes several performance deficits, as you indicated. Therefore, as described in Response 1, it is necessary to consider the change in performance when using high-intensity static stretching. Regarding this point, it is described in the Discussion section (Line 352-363).

Comment 14: 

Methods:

General- Unclear how many repetitions of each static stretch were undertaken? The authors do not mention the reliability of these measurements.

Response 14: 

Thank you for the comment. The participants received one set of high-intensity static stretching at different durations. We added following sentence.

“The participants received one set of stretching at each different duration.” (Line 178-179)

The previous study reported that the reliability of the measurements used in this study were acceptable (Hatano et al., Journal of Strength and Conditioning Research, 2017). We added the following sentence.

“The previous study reported that the reliability of the measurements used in this study was acceptable [8].” (Line 122-123)

Comment 15: 

Specific

Line 174 - 'Analysis' of variance

Response 15: 

Thank you for the comment. We revised it. (Line 185)

Comment 16:

Statistical analysis:

No mention of the calculation of effects via partial eta squared however they are presented in the results section.

Response 16: 

Thank you for the comment. The values of partial eta squared was calculated by using SPSS ver 25. We added the following sentence in the Statistical Analyses section.

“To describe the effect size, the partial eta squared value was calculated by using the SPSS software.” (Line 195-197)

Comment 17: 

No mention of correlations was made in this section however correlational results are presented (Lines 249-256)

Response 17: 

Thank you for the comment. We described the analyses of correlations (Line 192-194). Please confirm it.

Comment 18: 

Conclusions:

There could be more information surrounding the practical implications of the results.

Response 18: 

Thank you for the comment. In accordance with the comment, we added the practical implications in the Conclusions section.

“High-intensity SS for 10 seconds could be useful to increase flexibility and prevent injuries. However, athletes should use high-intensity SS for more than 15 seconds when they need great muscle strength immediately after stretching.” (Line 369-371)

 

Response to Reviewer #2

Comment 1: 

Introduction:

Line 45: “increases”.

Response 1: 

Thank you for the comment. We revised it. (Line 45)

Comment 2: 

Line 45: “that the passive property of the muscle-tendon unit”…

Response 2: 

Thank you for the comment. We revised it. (Line 45)

Comment 3: 

Lines 46-50: It’s not clear to me what the difference in the measurements reported here is. Please clarify.

Response 3: 

Thank you for the comment. To clarify the difference in the measurements reported here. We changed the text as follows.

From

“However, it is pointed out that the passive property of the muscle-tendon unit and tolerance for stretching influences ROM [5–7]. The passive property of the muscle-tendon unit is measured by using muscle-tendon unit stiffness, which is calculated from the torque-angle curve during passive joint movement [8–10]. On the other hand, tolerance for stretching is measured peak passive torque during passive joint movement [10–12].”

To

“However, it is pointed out that the passive property of the muscle-tendon unit and tolerance for stretching influences ROM [5–7]. The passive property of the muscle-tendon unit is measured by using muscle-tendon unit stiffness, which is calculated from the slope of the torque-angle curve during passive joint movement [8–10]. On the other hand, tolerance for stretching is measured by peak passive torque during passive joint movement [10–12]. ROM is defined as one at which the subject is able to have a maximally tolerable angle without pain. An increase in the passive torque during the passive joint movement indicated that the subjects received higher force without pain, which indicates that an increase of tolerance is obtained.” (Line 45-53)

Comment 4: 

The reference used to justify the link between muscle-tendon unit stiffness, and injury risk, is quite dated. In the last 20 years there has been a significant body of research that examines static stretching and injury risk, which seems to have been ignored here. It is important that this is addressed in the introduction.

Response 4: 

Thank you for the comment. In accordance with the comment, we added references to this section (Line 60)

Comment 5: 

Line 59: delete “it”

Response 5: 

Thank you for the comment. We revised it. (Line 62)

Comment 6: 

Line 69: should read “did not change” or “was not changed”

Response 6: 

Thank you for the comment. We revised it. (Line 72)

Comment 7: 

Line 83: I’m not sure that “demerit” is the word the authors are looking for here, but it's not clear to me what is meant.

Response 7: 

Thank you for the comment. We revised the text as follow.

“To minimize the pain during SS at high intensity, it is important to examine the optimal duration of the stretching.” (Line 87-89)

Comment 8: 

Materials and Methods:

The authors make a point of specifying that 20 sec was the stretching duration they wanted to assess. And this is the first mention of 10 sec and 15 sec. A justification for these durations in the introduction would be helpful.

Response 8: 

Thank you for the comment. We revised the Introduction sections as follow.

From

“The purpose of this study was to compare the duration of high-intensity SS (10, 15, and 20 seconds) on flexibility and strength in hamstrings and to clarify the optimal duration of SS at high intensity.”

To

“A previous study investigating static stretching protocol reported that SS for less than 10 seconds was rarely performed [1]. On the other hand, SS for 10- to 19 seconds was frequently used [1]. Therefore, the purpose of this study was to compare the duration of high-intensity SS (10, 15, and 20 seconds) on flexibility and strength in the hamstrings and to clarify the optimal duration of SS at high intensity.” (Line 90-94)

Comment 9: 

Mostly methods were very well reported. However, I’m a little unclear about how the stretch was obtained. What was the hip angle during stretching and testing? This will have a significant impact on the stretch that is felt at different angles of knee extension, but I can’t see this reported anywhere.

Response 9: 

Thank you for the comment. A previous study used in this measurement reported that the average angle of hip flexion was 111.2 ± 2.5 degrees. We added the following sentence to the Flexibility Assessment section.

“The previous study, which used the same assessment, reported that the average angle of hip flexion was 111.2 ± 2.5 degrees [8].” (Line 128-129)

Comment 10: 

Line 174: “one-way repeated measures analysis of variance”

Response 10: 

Thank you for the comment. We revised it. (Line 185)

Comment 11: 

Results:

I’m not sure about the need for Table 1. This is hardly a key result, and is not very informative, so this can be reported in the text. I’m not even sure the statistical analysis reported is necessary here, other than to compare the conditions during stretching.

Response 11: 

Thank you for the comment. In accordance with your comment, we reported the results of NRS in the text. (Line 251-258)

Comment 12: 

Discussion:

Paragraph starting Line 294 needs to be broken up. There are a number of ideas here, and this is hard to follow.

Response 12: 

Thank you for the comment. We divided the paragraph into two. The first paragraph discussed the results of the present study. The next paragraph discussed the sympathetic nerve.

Comment 13: 

Line 320: is a word missing here?

Response 13: 

Thank you for the comment. We revised the sentence. (Line 337)

Comment 14: 

Line 334: “person”

Response 14: 

Thank you for the comment. We revised it. (Line 351)

Comment 15: 

Line 338: is “fore” an extra work here?

Response 15: 

Thank you for the comment. We omitted the word. (Line 355)

---

## [Editor Report · Decision Letter 1]

22 Sep 2020

The optimal duration of high-intensity static stretching in hamstrings

PONE-D-20-18359R1

Dear Dr. Takeuchi,

We’re pleased to inform you that your manuscript has been judged scientifically suitable for publication and will be formally accepted for publication once it meets all outstanding technical requirements.

Kind regards,

Andrew Philip Lavender, PhD

Academic Editor

PLOS ONE

---

## [Editor Report · Acceptance letter]

25 Sep 2020

PONE-D-20-18359R1 

The optimal duration of high-intensity static stretching in hamstrings 

Dear Dr. Takeuchi:

I'm pleased to inform you that your manuscript has been deemed suitable for publication in PLOS ONE. Congratulations! Your manuscript is now with our production department. 

Kind regards, 

on behalf of

Dr. Andrew Philip Lavender 

Academic Editor

PLOS ONE